# Update in Molecular Testing for Intraocular Lymphoma

**DOI:** 10.3390/cancers14194546

**Published:** 2022-09-20

**Authors:** Michael J. Heiferman, Michael D. Yu, Prithvi Mruthyunjaya

**Affiliations:** 1Department of Ophthalmology and Visual Sciences, Illinois Eye and Ear Infirmary, University of Illinois at Chicago, Chicago, IL 60612, USA; 2Department of Ophthalmology, Byers Eye Institute, Stanford University, Palo Alto, CA 94303, USA

**Keywords:** vitreous biopsy, diagnostic vitrectomy, intraocular lymphoma, masquerade, ocular pathology

## Abstract

**Simple Summary:**

The diagnosis of primary vitreoretinal lymphoma and central nervous system lymphoma is challenging. Intraocular biopsy and molecular testing are important for the diagnosis of cases with intraocular involvement. Intraocular biopsy does not always result in a tissue diagnosis. There are many new molecular tests that are currently being used to improve the yield of intraocular biopsy. This article will review the available molecular tests for intraocular lymphoma.

**Abstract:**

The diagnosis of primary vitreoretinal lymphoma and central nervous system lymphoma is challenging. In cases with intraocular involvement, vitreous biopsy plays a pivotal role. Several diagnostic tests are employed to confirm a diagnosis and include cytologic evaluation, immunohistochemistry, flow cytometry, and cytokine analysis. The limitations of these conventional diagnostic tests stem from the often paucicellular nature of vitreous biopsy specimens and the fragility of malignant cells ex vivo. Several emerging molecular techniques show promise in improving the diagnostic yield of intraocular biopsy, possibly enabling more accurate and timely diagnoses. This article will review existing diagnostic modalities for intraocular lymphoma, with an emphasis on currently available molecular tests.

## 1. Introduction

The term intraocular lymphoma (IOL), previously known as reticulum cell sarcoma, refers to a heterogenous group of malignancies that can be subdivided based on tissue origin and histomorphology [1]. The most common IOL is primary vitreoretinal lymphoma, which is sometimes classified as a subtype of central nervous system (CNS) lymphoma due to their strong association [2].

Before pars plana vitrectomy was introduced in 1971, enucleation was necessary to make a tissue diagnosis of IOL [3,4,5]. Today, enucleation is reserved for patients without any possibility to restore vision or with intractable eye pain. With advancements in surgical techniques and diagnostic testing, such a drastic measure is rarely needed. Presently, the diagnosis of IOL depends on obtaining an intraocular biopsy, followed by a combination of cytologic evaluation, immunohistochemistry, and flow cytometry (Table 1).

The advent of polymerase chain reaction (PCR), however, has enabled easier methods for molecular testing further adding to the armamentarium of diagnostic modalities for IOL. The focus of this review will be on molecular testing of biopsy specimens for both primary vitreoretinal lymphoma and CNS lymphoma with intraocular involvement, which we will refer to as IOL.

A thorough medical history, ophthalmic examination, ophthalmic imaging, physical examination, and systemic ancillary testing should be performed prior to considering invasive procedures to diagnose IOL. Often patients with suspected IOL have failed various courses of immunosuppressive or antimicrobial therapy; this can delay the time to diagnosis [6]. Additionally, in patients with suspected IOL, a magnetic resonance image (MRI) of the brain and possibly a high-volume lumbar puncture with cytologic analysis should be performed due to the high incidence of CNS involvement [7]. Once the status of CNS involvement is determined, an ocular specimen is often obtained to make a tissue diagnosis.

Intraocular biopsy for the diagnosis of IOL requires careful planning with a multidisciplinary team including the patient, ophthalmologist, ocular pathologist, primary care physician, and a medical oncologist. The indication for biopsy and the clinical scenario should be reviewed with the ocular pathologist before the day of surgery to form a surgical plan, including the biopsy technique, specimens to be obtained, perioperative tissue handling, and tests to be performed. The biopsy technique chosen directly influences the type, integrity, and volume of the ocular specimen obtained. Since the volume of ocular tissue obtained is relatively small, optimizing the diagnostic yield is critical to decrease the need for a repeat surgical procedure. Biopsy specimen testing should prioritize studies based on clinical suspicion while keeping in mind the type, integrity, and amount of specimen needed for each test. The ophthalmologist and ocular pathologist should work together to involve cytology, microbiology, and send-out laboratory testing as needed. Intraocular biopsy is an invasive procedure with potential ocular morbidity, and its yield can be maximized by a strong relationship between the ophthalmologist and ocular pathologist [8].

## 2. Vitreous Biopsy

Vitreous biopsy plays a critical role in the diagnosis and management of IOL. The vitreous is the preferred location for obtaining a tissue specimen in patients with chronic vitreous cell of unknown etiology, especially in elderly patients partially responsive to immunomodulatory therapy. Vitrectomy can also aid in the diagnosis of IOL in cases with concurrent CNS disease on imaging and non-diagnostic cerebrospinal fluid analysis [9].

Vitreous biopsy specimens may not always result in the detection of neoplastic IOL cells. This is particularly common in cases with minimal vitreous findings, such as a very paucicellular vitreous cellular infiltration on slit lamp examination. Indeterminate results can also occur from cell degeneration due to a prolonged time between specimen collection and analysis. The quality of the specimen may also be insufficient to perform cytological or molecular analysis. Therefore, it may be necessary to perform a second vitrectomy with a chorioretinal biopsy or collect additional vitreous, but only if there are still vitreous cells remaining on examination.

## 3. Cytological Evaluation

The direct visualization of malignant cells by cytologic examination remains the current gold standard for diagnosis [10]. Accurate cytologic examination depends on the quality and integrity of vitreous samples, which are ideally collected by undiluted directed vitrectomy [11]. Subsequently, hematoxylin and eosin, Giemsa, and Diff-Quick stains are employed to identify malignant cells [12] (Table 1).

Over the years, various studies have demonstrated a range of diagnostic yield (10–90%) of vitreous fluid analysis to help establish a diagnosis of IOL [13,14,15,16,17,18,19,20,21]. This wide variation in reported diagnostic yield may be partially attributable to improvements in technique and testing strategies. The variable pre-test probability of suspected lymphoma in cases analyzed might also contribute, as lower diagnostic yields were reported in studies that included patients with diagnoses other than IOL.

However, cytologic evaluation has inherent limitations that might contribute to its variable diagnostic sensitivity as well (Table 2). Cytologic evaluation is frequently inconclusive, owing to the low cellularity of the vitreous fluid and the fragility of malignant cells during vitrectomy and ex vivo [19,22,23,24]. Cells that are successfully collected begin the process of apoptosis and morphological degradation within 60 min of vitreous aspiration [25]. Hence, vitreous specimens with a suspected IOL diagnosis should be handled expeditiously and placed in an appropriate transport media or fixative to preserve cytological details [26,27].

## 4. Immunohistochemistry and Flow Cytometry

The immunophenotype of monoclonality supports the cytological diagnosis of IOL and can be established using immunohistochemistry (IHC) or flow cytometry [41] (Table 1). IHC utilizes antibodies to stain specific protein markers of B-cells or T-cells on cytologic preparations, providing a semi-quantitative diagnostic adjunct to cytology. Most IOL cases are monoclonal B-cell lymphomas that stain positively for B-cell markers, including CD19, CD20, CD22, and CD79a, and show skewed expression of either immunoglobin kappa or lambda chain [42]. Germinal center markers such as BCL6 and CD10 can also be expressed [43]. Lineage determination can be coupled with Ki-67 staining to evaluate proliferative activity of lymphoma [44]. Rarely, IOL can be of T-cell origin and simulate a reactive inflammatory process; the lack of immunocytochemical markers for T-cell IOL, however, often makes the diagnosis of T-cell IOL more challenging [45].

Similar to IHC, flow cytometry entails tagging cells with fluorescent-labeled antibodies specific to cell-surface protein markers of interest. The cells are then suspended in a liquid phase and introduced into the flow cytometer, which activates each fluorescent tag separately and measures its light scatter to produce a distribution of cell immunophenotypes [46]. Compared to IHC, flow cytometry permits a more comprehensive and quantitative profiling of cell surface markers. Importantly, flow cytometry can simultaneously analyze several different markers and thereby characterize immunophenotype B- and T-cell subsets [26,47].

For IOL, flow cytometry-based diagnosis was first described in 1997 by Davis et al., who reported an improvement in IOL detection rate from 30% by cytology alone to 70% when coupled with flow cytometry [48]. More recent studies have shown flow cytometry to be comparable in sensitivity to cytokine analysis and immunoglobulin heavy chain (IgH) rearrangement analyses, with a reported sensitivity of 75–82% and specificity of 95–100% [30,31] (Table 2).

Despite its favorable diagnostic sensitivity and specificity, flow cytometry has several notable limitations. Up to 25% of B-cell lymphomas do not produce surface immunoglobulin light chains, thereby preventing cytometric recognition of these cells and reducing utility of the kappa:lambda ratio for clonality assessment [49]. As with other diagnostic techniques, when T-cells are identified, flow cytometry has a limited capacity to discriminate IOL from an inflammatory process; confirming the monoclonality of these T-cells would require molecular analysis. Flow cytometry also requires a minimum of around 100 cells, far more than for cytology, limiting its use in paucicellular ocular biopsies [50].

## 5. Cytokine Analysis (IL-6 and IL-10)

Cytokine analysis has become a useful adjunctive tool in differentiating IOL from inflammatory conditions, such as uveitis. B-cells secrete large amounts of interleukin-10 (IL-10), an immunosuppressive cytokine that promotes B-cell lymphoma cell proliferation [51,52]. In contrast, interleukin-6 (IL-6) is a pro-inflammatory cytokine commonly secreted into the vitreous by macrophages and T-cells under inflammatory conditions, including in the context of uveitis [53].

Cytokine analysis uses enzyme-linked immunosorbent assays (ELISAs) or cytokine multiplex assays to measure IL-10 and IL-6 levels in the vitreous and calculate the ratio of IL-10:IL-6 (Table 1). An IL-10:IL-6 ratio greater than 1.0 is suggestive of IOL [54,55]. Wolf et al. showed that an intravitreal IL-10:IL-6 >1.0 could correctly distinguish IOL from uveitis with approximately 75% sensitivity and specificity [32]. Subsequent studies have reported sensitivities of up to 90% with the IL-10:IL-6 ratio [22,28,33] (Table 2).

Measurement of IL-10 in the aqueous humor was also proposed as a screening test. Cassoux et al. reported average aqueous IL-10 levels of 543 picograms/milliliter (pg/mL) and 21.9 pg/mL in patients with IOL and uveitis, respectively, and found that a cutoff of 50 pg/mL in the aqueous could impart a sensitivity and specificity of approximately 90% [35].

However, the use of cytokine profiling is controversial for several reasons. Cytokine levels are particularly sensitive to corticosteroid and immunosuppressant therapy, leading to potentially misleading ratios. Furthermore, even in untreated eyes, an elevated IL-10:IL-6 ratio was reported in association with non-neoplastic uveitis [56]. Similarly, an IL-10:IL-6 ratio <1.0 was reported in biopsy-confirmed cases of IOL [23,28,57]. Availability of cytokine testing may be regional and institution specific, and finally, there is no clear standardization of detection methodology that influences repeatability.

## 6. Molecular Analysis

Molecular testing for IOL was largely enabled by the advent of polymerase chain reaction (PCR). Today, various forms of PCR, including allele-specific quantitative (AS PCR) and droplet digital (ddPCR), are used in combination with post-PCR analysis, such as Sanger sequencing and high-resolution melt (HRM) analyses, to identify established IOL biomarkers, such as B- and T-cell receptor clonality, B-cell lymphoma 2 (BCL2) translocation, and myeloid differentiation primary response 88 (MYD88) L265P mutation (Table 1).

Molecular testing carries several advantages over other conventional techniques. Compared to cytology, molecular testing does not require expert interpretation and is, thus, considered more objective. Molecular testing can also be performed on paucicellular samples, requiring a relatively low number of cells compared to those needed for cytology and flow cytometry. Furthermore, even archived samples, such as formalin-fixed paraffin-embedded tissue blocks, in which DNA quality is suboptimal, can undergo molecular testing [58,59].

### 6.1. B- and T-Cell Receptor Clonality

A molecular diagnosis of B- and T-cell lymphoma can be made by detection of clonal B-cell receptor (BCR) and T-cell receptor (TCR) gene rearrangements, respectively. The determination of the BCR clonality focuses on the immunoglobulin heavy chain (IgH), where typical gene rearrangements can contribute to an estimated receptor diversity of 10^12^ to 10^18^ [60,61]. This is achieved in part due to various combinations between four sets of defined gene segments: variable (V), diversity (D), joining (J), and constant (C). However, the majority of BCR diversity stems from the IgH-complementarity-determining region 3 (CDR3), which comprises a unique sequence spanning the junction of the rearranged V, D, and J segments [62]. Clonally expanded B-cell populations display identical copies of CDR3 at the junction of the rearranged VDJ segments [63]. The B-cells within inflammatory lesions, by comparison, contain diverse CD3 sequences within their IgH gene rearrangements. Analogous to the BCR, TCR diversity is also generated by both somatic VDJ recombination and the junctional region between these recombined gene segments, also termed CDR3 [64]. Normal TCR repertoire diversity is estimated at 10^11^ to 10^15^ [64].

To evaluate the diversity of CDR3 sequences in a sample of cells, consensus primers complementary to the conserved DNA flanking CDR3 regions are amplified by PCR. The products are then separated by electrophoresis or HRM analysis to determine the presence of a smear distribution (suggestive of an inflammatory lesion) or a single distinct band (suggestive of IOL).

The sensitivity of IgH gene analysis is generally high when combined with an optimized primer design and microdissection. Chan et al. demonstrated the importance of primer design in a study of 50 cytology-confirmed cases of IOL [65]. All 50 cases were subjected to microdissection, whereby relatively pure cell populations are selected from cytological slides to be used in downstream molecular analyses. This was followed by IgH gene analysis using three primer sets: framework region (FR) 3A, FR2A, and CDR3. The FR3A primer, which corresponds to a more conserved IgH segment and, thus, has better primer coverage, revealed positive rearrangements in 100% of cases, compared to 44% and 88% for the FR2A and CDR3 primers, respectively. Similarly, in 2011, Wang et al. yielded nearly 100% sensitivity and specificity when combining IgH and TCR gene analysis with microdissection [28] (Table 2). All 114 patients diagnosed with IOL—including 109 diagnosed with B-cell lymphoma and 5 with T-cell lymphoma—showed corresponding IgH or TCR gene rearrangements. Of the remaining 86 patients with uveitis, 85 demonstrated negative IgH or TCR gene rearrangements, resulting in only 1 false-positive outcome.

As reported by Chan et al. and others, false negatives can occur when clonal DNA within the CDR3 section is not detected due to insufficient or improper primer coverage and binding or poor DNA quality [66,67,68]. False positives, which are more common, typically result in cases of low cellularity, where a paucity of cells can mask true receptor diversity, resulting in a false appearance of clonality or pseudo-monoclonality [69]. Areas where compartmentalized B-cell expansion can occur, typically in response to infection or immunologic disorders, can also produce false positive results.

Despite these limitations, molecular testing based on receptor clonality offers a new depth of characterization with the potential to advance our understanding of IOL. For instance, Coupland et al. reported a case of oculocerebral lymphoma in which identical IgH clonality of cells derived from the eye and brain established the association between IOL and primary central nervous system lymphoma (PCNSL) [70].

### 6.2. B-Cell Lymphoma 2 (BCL2) Translocation

The B-cell lymphoma 2 (BCL2) gene on chromosome 18 encodes the bcl2 protein, which regulates apoptosis by inhibiting cells from programmed cell death. Translocation (14;18) leads to a juxtaposition of the BCL2 gene to the JH segments of the IgH locus on chromosome 14, leading to an overexpression of the bcl2 protein and enabling cells to overcome apoptosis [71,72,73]. Early studies reported BCL2/JH t(14;18) in 85–90% of cases of follicular lymphomas and only 30% of diffuse large B-cell lymphomas (DLBCL) of the germinal center B-cell-like (GBC) subtype [65,74,75,76]. BCL2 translocation in association with IOL was first reported by White et al. in 1993 and subsequently described in 57% of patients with IOL by Wallace et al. in 2006 [66,77].

Presently, BCL2 translocation status is not considered a diagnostic biomarker for IOL. However, studies of patients with DLBCL have suggested a potential prognostic value of BCL2 translocation. In an analysis of 106 DLBCL biopsy specimens, BCL2 translocation was reported in 27 (25.5%) cases and was found to be significantly associated with poorer overall survival and progression-free survival [76,78]. Kaminski et al. reported consistent findings, albeit with follicular lymphoma [79]. Other studies have reported a significantly younger patient age in association with BCL2 translocation, suggesting a role for the translocation to accelerate disease presentation and perhaps indicating a need for more aggressive therapy [70,73,74].

### 6.3. Myeloid Differentiation Primary Response 88 (MYD88)

Recently, sequencing studies have identified mutations in the MYD88 gene as an important, high-frequency oncogenic driver in B-cell lymphomas. The encoded MYD88 adaptor protein activates nuclear factor κ-light-chain-enhancer of activated B cells (NF-κB). Constitutive activation of NF-κB occurs when a single missense mutation at position 265 results in a substitution of leucine (L) to proline (P) in MYD88 (MYD88 L265P). MYD88 L265P and the resultant constitutive activation of NF-κB is central to activated B-cell-like-DLBCL pathogenesis but seems to play an especially critical role in IOL.

Based on a meta-analysis comprising 18 studies with 2002 total cases of DLBCL, mutations affecting MYD88, which were most commonly the canonical L265P mutation, were found in only 21% of DLBCL cases [80]. Stratification by subtype and location of DLBCL, however, revealed that MYD88 L265P was consistently higher in extranodal DLBCL in immune-privileged sites. Among these, vitreoretinal DLBCL ranked highest with a prevalence of 69–88% across numerous studies [37,38,40,80,81,82]. These data offer interesting insights into the pathogenesis of DLBCL and seem to suggest that B-cells must gain the MYD88 L265P mutation to survive and manifest at extranodal sites [83].

Of particular note, Bonzheim et al., in their MYD88 L265P analysis of 75 vitrectomy samples, detected MYD88 L265P, and thereby confirmed a diagnosis of IOL, in six cases initially reported as reactive or suspicious for lymphoma, based on cytology, IHC, and clonality analysis [37]. This increased the diagnostic sensitivity from 62 to 90% while maintaining diagnostic specificity [10,37] (Table 2). None of the cases classified as reactive tested positive for MYD88 mutations. To date, the numerous published reports on MYD88 L265P support its use as a molecular diagnostic adjunct for cases of IOL in which the cytology and IHC remain inconclusive [37,38,40,81,82,84].

MYD88 L265P mutation detection can be achieved through various molecular applications. Initial sequencing was performed using PCR with Sanger sequencing or HRM analysis. These were later replaced by AS PCR due to superior sensitivities [78,79]. Today, newer molecular techniques, such as droplet digital PCR (ddPCR), which partitions samples into nanoliter-sized droplets that are then amplified and enumerated independently, offer yet greater sensitivity [85,86]. Hiemcke-Jiwa et al. used ddPCR to analyze 12 paired vitreous and aqueous humor samples [36]. MYD88 L265P was detected in 9 of the 12 vitreous samples; of these, ddPCR detected the mutation in 8 of 9 corresponding aqueous samples. The high concordance rate (89%) between vitreous and aqueous samples underscores the ultrasensitivity of ddPCR.

Indeed, ddPCR was used to analyze cell-free DNA derived from serum in patients with confirmed PCNSL [87]. ddPCR successfully detected the mutation in 8 of 14 (57%) patients with confirmed MYD88 L265P based on tumor-derived DNA. Others analyzed cell-free circulating tumor DNA (ctDNA) using alternate techniques, including next-generation sequencing to assess for oncogenic mutations similar to MYD88 L265P. Bohers et al. and Kurtz et al., for instance, demonstrated noninvasive monitoring of DLBCL by analyzing ctDNA [88,89]. Such studies were not conducted for IOL and may prove more challenging due to presumably reduced circulating tumor DNA from an immune-privileged site. Nevertheless, with rapid improvements in ultrasensitive detection techniques, these non-invasive and increasingly reliable approaches in screening, diagnosis, and monitoring may soon be available for IOL.

## 7. Perioperative Tissue Handling

Here, we present our testing strategy in cases with the highest suspicion for IOL undergoing vitreous biopsy. Perioperative tissue handling should be clearly communicated between the ocular pathologist, ophthalmologist, operating room staff, and laboratory staff to maximize the yield of the biopsy procedure. The case should be reviewed with the ocular pathologist and cytopathologist prior to the day of surgery to form a perioperative plan including the biopsy technique, specimens to be obtained, tissue handling, and tests to be performed. Outside pathology consultation should be used if an experienced ocular pathologist is not available at the treating hospital.

An undiluted vitreous biopsy specimen should be obtained with a goal volume of 2 mL. This undiluted specimen is divided evenly into three parts for cytology, flow cytometry, and molecular testing. The cytology specimen is placed in Cytolyt (Cytyc Corporation, Marlborough, MA, USA) to preserve cell morphology. The flow cytometry specimen is placed in Roswell Park Memorial Institute (RPMI) culture media to prevent cell death. A diluted vitreous biopsy specimen is obtained with a volume of 10 mL. This diluted specimen is divided evenly into three parts for microbiology, cytology, and flow cytometry. The microbiology specimen is sent for aerobe, anaerobe, and fungal culture with or without viral PCR depending on the level of suspicion. PCR for conserved 16S and 18S ribosomal RNA gene sequences of bacteria and fungi, respectively, can also be performed on this specimen. The vitrector cartridge can also be sent for microbiology and PCR at the end of the case. The vitreous specimens are brought to the laboratory by a reliable team member and distributed for testing according to the preoperative plan made with the ocular pathologist.

## 8. Conclusions

The diagnosis of IOL is often challenging given its propensity to masquerade as other ocular diseases. While intraocular biopsy with cytological examination remains the mainstay of IOL diagnosis, molecular testing is emerging in an increasingly important adjunctive role. Molecular testing has the potential to not only improve diagnostic yield but also extract additional disease characterizing and prognostic information at a depth unafforded by cytological examination alone [90]. Genomic analysis of IOL is actively being pursued and has already revealed novel therapeutic targets towards the goal of personalized management of patients with IOL [91,92,93]. Ongoing proteomic analysis can identify novel diagnostic biomarkers for IOL [94]. Further advancements in this area have the potential to improve timely diagnoses of IOL, improving patient outcomes.

## Figures and Tables

**Table 1 cancers-14-04546-t001:** Diagnostic Tests for Intraocular Lymphoma.

Test	Analysis
Cytology	H&E, Giemsa, Diff-Quik
Immunohistochemistry and Flow Cytometry	CD19, CD20, CD22, CD79a
	BCL6, CD10
	Ki-67
	Rarely T-cell markers
Cytokine analysis	IL-10:IL-6 ratio >1.0
	Aqueous IL-10 level
Polymerase chain reaction	B- and T-cell receptor clonality
	B-cell lymphoma 2 (BCL2) translocation
	Myeloid differentiation primary response 88 (MYD88) L265P mutation

**Table 2 cancers-14-04546-t002:** Performance of Diagnostic Tests for Intraocular Lymphoma.

Test	Sensitivity	Specificity	References
Cytology	31–87%	98–100%	[14,16,17,19,28,29]
Flow Cytometry	82–100%	95–100%	[17,30,31]
IL-10:IL-6 ratio >1.0	75–90%	75–100%	[17,22,28,31,32,33,34]
Aqueous IL-10 level	90%	90%	[35]
Immunoglobulin heavy chain (IgH) rearrangement	40–100%	79–99%	[17,19,28,34]
Myeloid differentiation primary response 88 (MYD88) L265P mutation	67–91%	92–100%	[28,36,37,38,39,40]

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
