# Peer review of "Update in Molecular Testing for Intraocular Lymphoma"

_cancers, 2022, doi:10.3390/cancers14194546_

Round 1
Reviewer 1 Report
The article provides a comprehensive review of the current conventional and new technologies, methods and advances in the diagnosis of ocular lymphoma, which will help the readers to get a wide view of the current status of the diagnosis of this disease, which will help the reader to get a relatively clear view of the current status of the diagnosis of this disease.
It is recommended that more newly published papers should be cited in the manuscript, especially researches concerning sequencing and omics be applied in the text.
This article provides a comprehensive review of the current conventional and new technologies, new methods, and new advances in the diagnosis of ocular coloboma, which will help the reader to get a relatively clear view of the current state of the diagnosis of this disease, with some guiding value.
The inadequacy is that the cited references are older, especially less new literature has been published in recent years, and it is recommended to supplement the research results of sequencing, omics in the text.
Reviewer 2 Report
This is a very useful and well written review.
I recommend the following minor text edits for clarity:
1. Table 1 is only referred to in the introduction. As the terms and acronym's used in the table are only fully defined later in the main text, it would be helpful to the reader to include further references back to Table 1 where these terms (tests) are defined and discussed in the relevant sections.
Suggested locations for further references to Table 1 listed below; ie, for inclusion in parentheses at the end of a sentence "...(Table 1).".
[Or, the authors may choose to include references to Table 1 within the sentence structure itself, such as "...as summarized in table 1.."].
Line 85. End of last sentence in 1st paragraph on cytological evaluation tests.
Line 102. End of first sentence in 1st paragraph on IHC and flow cytometry.
Line 143. End of first sentence in the 2nd paragraph of cytokine analysis section.
Line 165. End of last sentence in first paragraph for section on molecular testing.
2. Text edits around use of acronym's in main text listed below:
Line 48. Include full term for acronym 'MRI' ahead of its first use.
Line 84. As 'Diff-Quick' is an abbreviation, suggest defining term in full ahead of use of abbreviation in the main text (particularly as this abbreviation also used in Table 1). ie; "...differential quick ('Diff-Quick') stains..."
Line 160-161. Full definition for PCR not necessarily required here as defined earlier in main text.
Line 292. Define millilitres (mL) at first use.
